

# Explaining the convector effect in canopy turbulence by means of large-eddy simulation

Tirtha Banerjee[1], Frederik De Roo[1], and Matthias Mauder[1]

[1]Karlsruhe Institute of Technology (KIT) Institute of Meteorology and Climate Research, Atmospheric Environmental Research (IMKIFU), D-82467 Garmisch-Partenkirchen, Germany

*Correspondence to:* Tirtha Banerjee (tirtha.banerjee@kit.edu, banerjeetirtha10@gmail.com)

**Abstract.** Semi-arid forests are found to sustain a massive sensible heat flux in spite of having a low surface to air temperature difference by lowering the aerodynamic resistance to heat transfer ($r_H$) – a property called 'canopy convector effect' (CCE). In this work large-eddy simulations are used to demonstrate that CCE appears more generally in canopy turbulence. It is indeed a generic feature of canopy turbulence: $r_H$ of a canopy is found to reduce with increasing unstable stratification, which effectively increases the aerodynamic roughness for the same physical roughness of the canopy. This relation offers a sufficient condition to construct a general description of CCE. In addition, we review existing parameterizations for $r_H$ from the evapotranspiration literature and test to what extent they are able to capture the CCE, thereby exploring the possibility of an improved parameterization.

## 1 Introduction

Understanding the role of turbulence in interactions between vegetation canopies and the atmosphere is crucial to interpret momentum and scalar fluxes above vegetation. This is relevant for a number of practical applications such as regional and global weather and climate modeling, energy balance closure studies, developing forest management strategies etc. Measurement campaign networks such as FLUXNET monitor carbon, water and energy fluxes on a long term basis for this same reason (Baldocchi et al., 2001) to study how different ecosystems interact with the atmosphere and influence local and global weather and climate. One such measurement campaign (Rotenberg and Yakir, 2011) focused on semi-arid ecosystems, specifically the Yatir forest situated in the Negev desert in Israel, to study the survival and productivity of the pine forest in spite of the high radiation load and suppressed latent heat flux. An important outcome of this campaign was the concept of 'canopy convector effect' (CCE) introduced by Rotenberg and Yakir (2010). To quote them (Rotenberg and Yakir, 2010), "With suppressed latent heat flux (LE) because of lack of water, the forest is transformed into an effective 'convector' that exploits the low tree density and open canopy and, consequently, high canopy-atmosphere aerodynamic coupling." Rotenberg and Yakir (2010) ascribed the origin of the CCE to the roughness difference between desert and forest. However, in the present work, we demonstrate that the canopy convector effect appears more generally in canopy turbulence. In fact, we show that the CCE is also a generic artifact of homogeneous canopy turbulence by using large-eddy simulations (LES). In doing so, the canopy aerodynamic resistance to heat transfer is revisited. The canopy aerodynamic resistance is a concept borrowed from the evapotranspiration literature





where it represents the resistance between the idealized 'big-leaf' (a reduced order representation of the fully heterogeneous three dimensional canopy) and the atmosphere for heat or vapor transfer (Monteith, 1973; Foken et al., 1995; Alves et al., 1998; Monteith and Unsworth, 2007). We investigate if the existing parameterizations of the canopy aerodynamic resistance exhibit CCE and we identify uncertainties in their application. As the CCE is the crucial mechanism that ensures the survival

of the Yatir forest, an improved physical understanding of the CCE is of primordial importance when considering large-scale afforestation in semi-arid regions.

## 2   Background and theory

### 2.1   The canopy convector effect and aerodynamic resistance

As mentioned earlier, the canopy convector effect was introduced by Rotenberg and Yakir (2010, 2011) while studying the

interaction of vegetation cover with the surface radiation balance for the Yatir forest. The annual average incoming solar radiation in the Yatir forest is about $238\,\mathrm{Wm^{-2}}$ comparable to that in the Sahara desert but the net radiation ($R_n$) is about $35\%$ higher than the Sahara (Rotenberg and Yakir, 2010) due to the lower albedo of the forest. However, both remote sensing and local measurements indicated that the surface temperature of the forest canopy in Yatir is lower than the surface temperature of the nearby non-forested area – on annual average by about $5\mathrm{K}$. This is striking, as firstly, the lower albedo (by 0.1) of the

forest than that of the surrounding shrubland translates into an approximate $24\,\mathrm{Wm^{-2}}$ increase of radiation load on the forest canopy. Secondly, the cooler canopy surface suppresses the upwelling longwave radiation, resulting in an additional increase of radiation load by about $25\,\mathrm{Wm^{-2}}$. The combined annual increase of radiation load by about $50\,\mathrm{Wm^{-2}}$ associated with the Yatir afforestation in the Negev is quite high and is comparable to net annual radiation difference between the Sahara desert and Denmark, for example (Rotenberg and Yakir, 2010). Thirdly, the latent heat flux of evapotranspiration ($LE$), the obvious

cooling and energy dissipation mechanism in temperate forests, is not an option since water is virtually unavailable for about 7 months a year. Thus sensible heat flux ($H$) is the only major heat dissipation route, translating into a Bowen ratio ($H/LE$) as high as 20 or more — unlike temperate forests with Bowen ratio $\approx 1$. In the Yatir forest, the entire net solar radiation flux (up to $800\,\mathrm{Wm^{-2}}$) is equilibrated by a massive sensible heat flux ($H$) of similar magnitude. Note that this high $H$ cannot be explained by the difference between surface and air temperature ($\Delta T = T_s - T_a$) as the canopy surface is cooler than the

surrounding desert surface in this case, but the air temperatures above desert and forest canopy are similar. To expound this apparent contradiction of larger sensible heat flux for smaller $\Delta T$, it is important to recall that

$$H = -\rho C_p \frac{T_a - T_s}{r_H},\qquad(1)$$

where $\rho$ and $C_p$ are the density and specific heat capacity of air respectively, $T_a$ is air temperature, $T_s$ is canopy surface temperature and $r_H$ is the canopy aerodynamic resistance to heat transfer. Hence the large $H$ is not explained by the temperature

difference ($\Delta_T$) but by a decreased $r_H$. Thus the semi-arid forest with its low tree density and large surface area becomes an efficient low aerodynamic resistance 'convector' that is well coupled to the atmosphere above (Rotenberg and Yakir, 2010, 2011). This 'canopy convector effect' (CCE) is sufficiently efficient to support the massive sensible heat flux larger than the





surrounding Negev desert, still maintaining a relatively cooler (than the desert) surface temperature (of the canopy top). It is worth noting here that equation 1 offers a very simplistic description of the complex mixing process in the surface layer, however, it should be interpreted as a zeroth order representation of the corresponding processes. Rotenberg and Yakir (2010) identified the difference of roughness between the desert and forest as the underlying mechanism of CCE by arguing that

$r_H \propto 1/PAI$ where $PAI$ denotes the plant area index. However, in this work, we attempt to identify a more fundamental mechanism behind CCE which is more strongly connected to the feature of canopy turbulence. Therefore we hypothesize that even with the same physical roughness, variation of the aerodynamic roughness is a sufficient condition for displaying CCE. This difference of aerodynamic roughness for the same physical roughness (of the same vegetation canopy) can be generated by changing the intensity of atmospheric stratification (Zilitinkevich et al., 2008). Thus observing the variation of the canopy

aerodynamic resistance to heat transfer ($r_H$) with atmospheric instability is a sufficient condition to demonstrate the generality of the CCE. To be more precise, if $r_H$ is found to decrease with increasing unstable stratification, that would exhibit the fact that canopy turbulence effectively reduces the aerodynamic resistance to cope with heat stressed environments, i.e the canopy convector effect would manifest itself. LES provides a useful and meanwhile standard tool for studying canopy turbulence under different conditions of atmospheric stratification. A recent publication by Patton et al. (2015) studied the influence of

different atmospheric (in)stability classes on coupled boundary layer-canopy turbulence. In this work, those same (in)stability classes are simulated to put our hypothesis to the test.

### 2.2 Parameterizations for canopy aerodynamic resistance to heat transfer

Apart from the LES outcomes, it is also important to study if the existing parameterizations of $r_H$ can exhibit CCE. Parameterizations of $r_H$ in the literature use Monin-Obukhov similarity theory (MOST) extensively. MOST can provide corrections for

the vertical profile of the mean longitudinal velocity $u$ and potential temperature ($T_a - T_s$) under thermal stratification, which deviates from the traditional log-law under neutral conditions. Thus under MOST, with the assumption that the vegetation is low, dense and horizontally homogeneous,

$$u = \frac{u_*}{\kappa}\left[\ln\left(\frac{z-d}{z_{0m}}\right) - \psi_m\left(\zeta, \zeta_{0m}\right)\right], \tag{2}$$

and

$$T_a - T_s = P_{r0}\frac{T_*}{\kappa}\left[\ln\left(\frac{z-d}{z_{0h}}\right) - \psi_h\left(\zeta, \zeta_{0h}\right)\right], \tag{3}$$

where $u_*$ is the friction velocity, $\kappa$ is von Kármán constant, $z$ is height from the ground, $d$ is the zero-plane displacement height taken as $(2/3)h_c$ as per literature (Seginer, 1974; Shuttleworth and Gurney, 1990; Alves et al., 1998), and $\zeta = (z-d)/L$ is called the stability parameter. $L$ is the Obukhov length, computed as

$$L = -\frac{u_*^3 T_a}{\kappa\, g\overline{w'T'}}, \tag{4}$$

where $g = 9.81\mathrm{ms}^{-2}$, the gravitational acceleration. $\overline{w'T'}$ is the sensible heat flux — assumed to be constant in the surface layer (Foken, 2006). Negative $\zeta$ indicates unstable stratification and thus $\zeta$ decreases with increasing instability. $z_{0m}$ and $z_{0h}$





are the characteristic roughness lengths for momentum and heat transfer respectively. $\zeta_{0m} = z_{0m}/L$ and $\zeta_{0h} = z_{0h}/L$ are the stability parameters associated with roughness lengths. $P_{r0} = K_m/K_h$ is the turbulent Prandtl number where $K_m$ and $K_h$ are eddy diffusivities of momentum and heat, respectively. $T_*$ is a characteristic temperature scale, obtained from $H$ and the characteristic velocity scale, i.e.,

$$H = -\rho C_p u_* T_*. \tag{5}$$

Combining equations 1, 2, 3 and 5, one can write

$$r_H = \frac{P_{r0}}{\kappa^2 u} \left[ \ln\left(\frac{z-d}{z_{0m}}\right) - \psi_m\left(\zeta, \zeta_{0m}\right) \right] \left[ \ln\left(\frac{z-d}{z_{0h}}\right) - \psi_h\left(\zeta, \zeta_{0h}\right) \right], \tag{6}$$

where $\psi_m$ and $\psi_h$ are the integral stability correction functions for momentum and heat, respectively. Following Liu et al. (2007), they can be parameterized for unstable conditions as (Dyer and Hicks, 1970; Paulson, 1970; Dyer, 1974; Garratt, 1977; Webb, 1982)

$$\psi_m\left(\zeta, \zeta_{0m}\right) = 2\ln\left(\frac{1+x}{1+x_0}\right) + \ln\left(\frac{1+x^2}{1+x_0^2}\right) - 2\tan^{-1}x + 2\tan^{-1}x_0, \tag{7}$$

$$\psi_h\left(\zeta, \zeta_{0h}\right) = 2\ln\left(\frac{1+y}{1+y_0}\right); \tag{8}$$

where $x = (1-\gamma_m\zeta)^{1/4}$, $x_0 = (1-\gamma_m\zeta_{0m})^{1/4}$, $y = (1-\gamma_h\zeta)^{1/2}$ and $y_0 = (1-\gamma_h\zeta_{0h})^{1/2}$. Different values for the parameters $\gamma_m$ and $\gamma_h$ are reported in the literature, and the ones suggested by Paulson (1970) are used, i.e., $\gamma_m = \gamma_h = 16$. This formulation for $r_H$ given by equation 6 with some approximations ($\zeta_{0m} = \zeta_{0h} = 0$) was first used by Thom (1975) and is called the 'reference parameterization' (Liu et al., 2007). The full form of equation 6 was used by Yang et al. (2001) with their only approximation being $P_{r0} = 1$. Several other studies also used semi-empirical and empirical parameterizations and included the bulk Richardson number $Ri_B$ (Monteith, 1973) given by

$$Ri_B = \frac{g}{T_a} \frac{(T_a - T_s)(z-d)}{U_\parallel^2}, \tag{9}$$

with $U_\parallel$ the horizontal wind speed at the height that corresponds to the $T_a$ measurement.

Liu et al. (2007) compiled different parameterizations of $r_H$ which we will test in the context of the canopy convector effect against our LES output. Table 1 lists the details of the different parameterizations as compiled by Liu et al. (2007). These parameterizations based on MOST (Thom, 1975; Yang et al., 2001), empirical (E) (Verma et al., 1976; Hatfield et al., 1983; Mahrt and Ek, 1984; Xie, 1988) and semi-empirical (SE) (Choudhury et al., 1986; Viney, 1991) assumptions can be classified into two categories. Formulations by Thom (1975), Choudhury et al. (1986), Yang et al. (2001) and Viney (1991) have assumed $z_{0m} \neq z_{0h}$, which should be a more realistic assumption. On the other hand, formulations by Verma et al. (1976), Hatfield et al. (1983), Mahrt and Ek (1984) and Xie (1988) assumed $z_{0m} = z_{0h}$. Different parameters used in the empirical formulations are also listed in table 1.





**Table 1.** Different parameterizations of $r_H$ as compiled by Liu et al. (2007).

| Source | Parameterization of $r_H$ | Coefficients | Assumption |
|---|---|---|---|
| Thom (1975) | $r_H = \dfrac{1}{\kappa^2 u}\left[\ln\left(\dfrac{z-d}{z_{0m}}\right) - \psi_m(\zeta)\right]\left[\ln\left(\dfrac{z-d}{z_{0h}}\right) - \psi_h(\zeta)\right]$ | $\zeta_{0m} = \zeta_{0h} = 0$ | $z_{0m} \neq z_{0h}$, MOST |
| Yang et al. (2001) | $r_H = \dfrac{1}{\kappa^2 u}\left[\ln\left(\dfrac{z-d}{z_{0m}}\right) - \psi_m(\zeta, \zeta_{0m})\right]\left[\ln\left(\dfrac{z-d}{z_{0h}}\right) - \psi_h(\zeta, \zeta_{0h})\right]$ | NA | $z_{0m} \neq z_{0h}$, MOST |
| Choudhury et al. (1986) | $r_H = \dfrac{1}{\kappa^2 u}\left[\ln\left(\dfrac{z-d}{z_{0m}}\right)\right]\left[\ln\left(\dfrac{z-d}{z_{0h}}\right)\right](1 - \beta Ri_B)^{-3/4}$ | $\beta = 5$ | $z_{0m} \neq z_{0h}$, SE |
| Viney (1991) | $r_H = \dfrac{1}{\kappa^2 u}\left[\ln\left(\dfrac{z-d}{z_{0m}}\right)\right]\left[\ln\left(\dfrac{z-d}{z_{0h}}\right)\right][a + b(-Ri_B)^c]^{-1}$ | $a,b,c = f((z-d)/z_{0m})$ | $z_{0m} \neq z_{0h}$, SE |
| | $a = 1.0591 - 0.0552\ln\left(1.72 + \left[4.03 - \ln\left(\frac{z-d}{z_{0m}}\right)\right]^2\right)$ $b = 1.9117 - 0.2237\ln\left(1.86 + \left[2.12 - \ln\left(\frac{z-d}{z_{0m}}\right)\right]^2\right)$ $c = 0.8437 - 0.1243\ln\left(3.49 + \left[2.79 - \ln\left(\frac{z-d}{z_{0m}}\right)\right]^2\right)$ | | |
| Verma et al. (1976) | $r_H = \dfrac{1}{\kappa^2 u}\left[\ln\left(\dfrac{z-d}{z_{0m}}\right)\right]^2(1 - 16Ri_B)^{-1/4}$ | NA | $z_{0m} = z_{0h}$, E |
| Hatfield et al. (1983) | $r_H = \dfrac{1}{\kappa^2 u}\left[\ln\left(\dfrac{z-d}{z_{0m}}\right)\right]^2(1 + \beta Ri_B)$ | $\beta = 5$ | $z_{0m} = z_{0h}$, E |
| Mahrt and Ek (1984) | $r_H = \dfrac{1}{\kappa^2 u}\left[\ln\left(\dfrac{z-d}{z_{0m}}\right)\right]^2\left[\dfrac{1 + c(-Ri_B)^{1/2}}{1 + c(-Ri_B)^{1/2} - 15Ri_B}\right]$ | $c = \dfrac{75\kappa^2\left(\frac{z-d+z_{0m}}{z_{0m}}\right)^{1/2}}{\left[\ln\left(\frac{z-d+z_{0m}}{z_{0m}}\right)\right]^2}$ | $z_{0m} = z_{0h}$, E |
| Xie (1988) | $r_H = \dfrac{1}{\kappa^2 u}\left[\ln\left(\dfrac{z-d}{z_{0m}}\right)\right]^2\left[1 + \dfrac{\left[1 - 16Ri_B\ln\left(\frac{z-d}{z_{0m}}\right)\right]^{-1/2}}{\ln\left(\frac{z-d}{z_{0m}}\right)}\right]$ | NA | $z_{0m} = z_{0h}$, E |





One important point to notice is that only the formulation by Yang et al. (2001) uses the stability parameters associated with the roughness lengths $\zeta_{0m}$ and $\zeta_{0h}$. Also note that all parameterizations assume a turbulent Prandtl number of unity, i.e., the diffusivities for momentum and heat are assumed to be the same. We shall later discuss the consequence of letting this parameter vary. Another important approximation necessary to evaluate all formulations in table 1 is a prescription for the roughness

lengths $z_{0m}$ and $z_{0h}$. Effects of different roughness lengths will be investigated in the following section. However, a relation between the two roughness lengths ($\kappa B^{-1} = \ln(z_{m0}/z_{0h})$) was proposed by Owen and Thomson (1963) and Chamberlain (1968), where $\kappa B^{-1}$ is called an 'excess resistance parameter'. Yang et al. (2001) suggested an average value of $\kappa B^{-1} = 2.0$ (Liu et al., 2007), which will be used throughout this work.

## 3 Methodology

The PALM large-eddy simulation model (Raasch and Schröter, 2001; Maronga et al., 2015) is used to investigate this generic nature of canopy convector effect. The representation of the canopy in the LES follows the standard distributed drag parameterization (Shaw and Schumann, 1992; Watanabe, 2004; Patton et al., 2015) by adding an additional term in the momentum budget equations as $F_{d_i} = -C_d a |\mathbf{u}| u_i$ where $a$ is a one sided frontal plant area density (PAD), $C_d$ is a dimensionless drag coefficient assumed to be 0.3 (Katul et al., 2004; Banerjee et al., 2013), $|\mathbf{u}|$ is the wind speed and $u_i$ is the corresponding

velocity component ($i = 1, 2, 3$, i.e. $u$, $v$ and $w$). The effect of the canopy on the subgrid scale (SGS) turbulence is accounted for by adding a sink term to the prognostic equation for the SGS turbulent kinetic energy ($e$) as $F_\epsilon = -2C_d a |\mathbf{u}| e$. For closure of the SGS covariance terms, PALM uses the 1.5 order closure developed by Deardorff (1980) as modified by Moeng and Wyngaard (1988) and Saiki et al. (2000), which assumes a gradient-diffusion parameterization. The diffusivities associated with this gradient-diffusion are parameterized using the subgrid-scale turbulent kinetic energy (SGS-TKE) and includes a prognos-

tic equation for the SGS-TKE. This SGS-TKE scheme after Deardorff (1980) is deemed to be an improvement over the more traditional Smagorinsky (1963) parameterization since the SGS-TKE allows for a much better estimation for the velocity scale corresponding to the subgrid-scale fluctuations (Maronga et al., 2015). Further details of the LES model can be found in the literature and are not discussed here (Shaw and Schumann, 1992; Watanabe, 2004; Maronga et al., 2015; Patton et al., 2015). For our simulation the number of grid points in the $x$, $y$ and $z$ directions are 320, 320 and 640 respectively, with grid resolution

of 3.91 m, 3.91 m and 1.95 m in the respective direction. Each simulation has a simulated time of 10000 s with a time step of 0.1 s, while the output of first 6400 s are discarded before achieving computational quasi-equilibrium. The canopy height ($h_c$) is taken as 35.0 m with a plant area index (PAI) of 5.0. It is important to note that Rotenberg and Yakir (2011) reported an effective PAI of about 5–6 for heat exchange for the Yatir forest. This makes our PAI similar to a recent simulation study of Dias-Junior et al. (2015). In fact, as we already simulate a homogeneous canopy to show that the CCE appears more generically above

vegetation canopies, we have decided to tailor our simulations following the examples of Patton et al. (2015) and Dias-Junior et al. (2015) in order to allow a better comparison of the LES data. The vertical distribution of plant area density ($a$) follows the pdf of a Beta distribution as described in Markkanen et al. (2003) and the parameters $\alpha$ and $\beta$ controlling the vertical distribution of foliage are set as 3.0 and 2.0 respectively to simulate a PAD distribution similar to Dias-Junior et al. (2015). The





**Table 2.** Parameters to drive the simulations for five different (in)stability classes namely near-neutral (NN), weakly unstable (WU), moderately unstable (MU), strongly unstable (SU) and free convection (FC) are similar to Patton et al. (2015). $U_g$ and $V_g$ denote geostrophic wind speeds, $\overline{w'T'}_s$ denotes ground surface sensible heat flux, $T_s$ denotes ground surface potential temperature and $q_s$ denotes specific humidity at the ground surface.

| Stability class | $(U_g, V_g)\,(ms^{-1})$ | $\overline{w'T'}_s\,(Kms^{-1})$ | $T_s\,(K)$ | $q_s\,(g/g)$ |
|:---:|:---:|:---:|:---:|:---:|
| NN | 20, 0 | 0.24 | 307.7 | 0.02 |
| WU | 10, 0 | 0.21 | 307.7 | 0.02 |
| MU | 5, 0 | 0.20 | 307.7 | 0.02 |
| SU | 2, 0 | 0.20 | 307.7 | 0.02 |
| FC | 0, 0 | 0.23 | 307.7 | 0.02 |

parameters to drive the simulations for five different (in)stability classes namely near-neutral (NN), weakly unstable (WU), moderately unstable (MU), strongly unstable (SU) and free convection (FC) are similar to those of Patton et al. (2015) and are presented in table 2. Note that the canopy convector effect as a general phenomenon should not depend on water content in the soil-plant-atmosphere continuum and moreover, the PALM-LES does not take into account any physiological processes which normally happen with a larger time scale. Nevertheless, instead of simulating a specific dry water free environment, some moisture at the lower surface is provided and the boundary conditions for surface moisture content are taken similar to the simulations of Dias-Junior et al. (2015) as well. The initial conditions of the potential temperature (and moisture) profile as also taken similar to Dias-Junior et al. (2015). Another important point to note is that instead of lowering the wind speeds while maintaining similar sensible heat fluxes, the different stability classes can also be achieved by maintaining the same wind speed and ramping up the surface sensible heat fluxes. However, this should not affect the generic feature of CCE as discussed at the end of section 2.1.

## 4 Results and Discussions

### 4.1 Comparison with LES

The results of the LES simulations are presented in figure 1 as temporally and spatially averaged vertical profiles for all five stability classes, where the lightest cyan shade indicates near neutral and the most magenta shade indicates free convective conditions. Panel (a) shows the mean wind speed ($U$), panel (b) shows the standard deviation of longitudinal velocity fluctuations ($\sigma_u$) and panel (c) shows the friction velocity (which can be taken as a measure of turbulent intensity) ($u_*$) at every level for each simulation

$$u_* = \left( \overline{u'w'}^2 + \overline{v'w'}^2 \right)^{1/4}. \tag{10}$$

In the second row, panel (d) shows profiles of temporally and spatially averaged potential temperature ($T$), panel (e) shows the kinematic sensible heat flux ($\overline{w'T'}$) and panel (f) shows the Prandtl number $P_{r0} = K_m/K_h$. The profiles (except $P_{r0}$) are




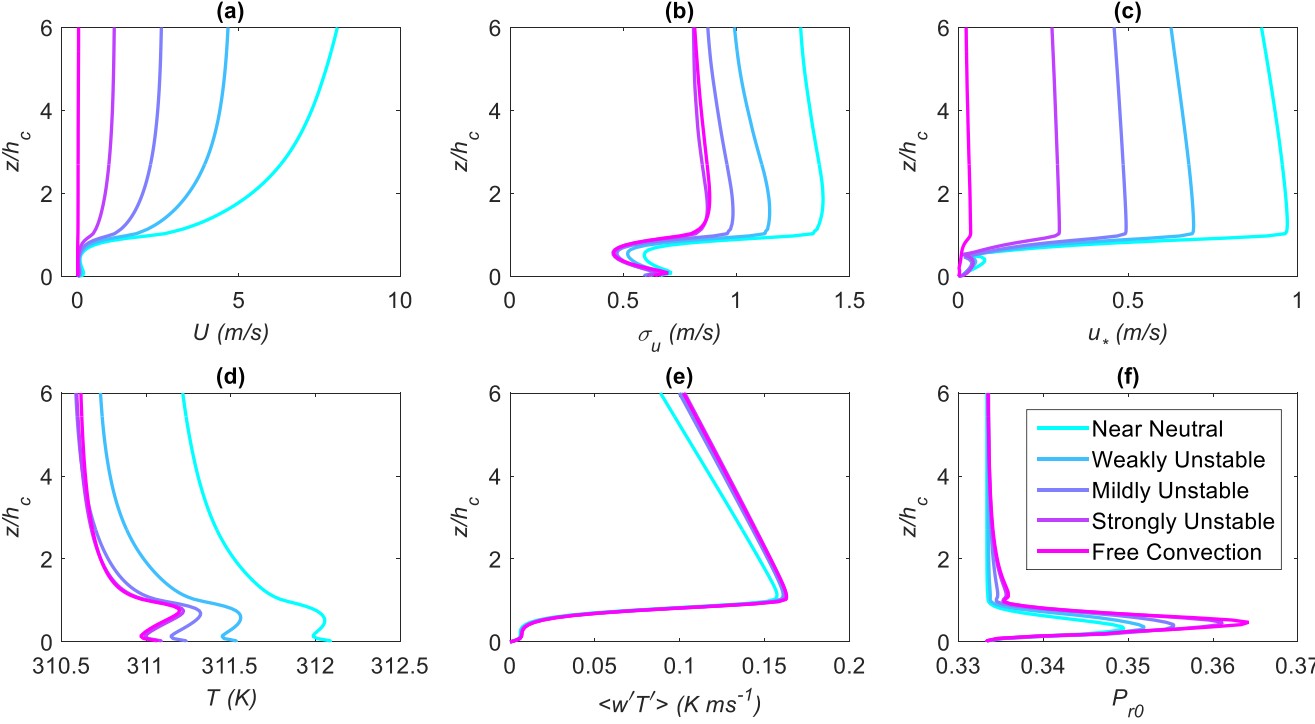

**Figure 1.** Summary statistics of five LES simulations showing the variations between different stability classes in increasing order of instability- from near neutral to free convection color coded as indicated in the legend.

shown in their dimensional form to clearly illustrate the differences between the different stability conditions. The simulation results closely follow the results presented in Patton et al. (2015) and Dias-Junior et al. (2015). It is interesting to observe that the magnitude of velocity, the velocity fluctuations and the turbulent intensity decreases gradually from the near neutral to free convective conditions, i.e., with increasing instability. The potential temperature also reduces with increasing instability at all

5    heights. On the other hand, the sensible heat flux appears to increase with increasing stability, especially more above the forest ($z/h_c = 1$ indicates the canopy top).

Figure 2 shows temporally and spatially averaged vertical profiles for the different stability conditions with the same color coding as figure 1. We investigate the vertical profile of $r_H$ in order to assess the uncertainty that arises from varying the reference height for the air temperature under varying stability. The temperature of the canopy top is taken as the surface

10    temperature ($T_s$) and thus results are shown from above the canopy top, i.e., $z/h_c = 1$. Panel (a) shows the difference of surface and air temperature ($T_s - T_a(z)$). Panel (b) shows the stability parameter $\zeta$ at every level computed as $\zeta = (z-d)/L$ as explained in section 2. Panel (c) plots canopy aerodynamic resistance to heat transfer ($r_H$) at every level computed from eq. 1. As evident from panel (c), the aerodynamic resistance reduces with increasing instability, confirming the hypothesis constructed earlier and thus clearly demonstrating the canopy convector effect (CCE). As noted by Zilitinkevich et al. (2008), with increasing





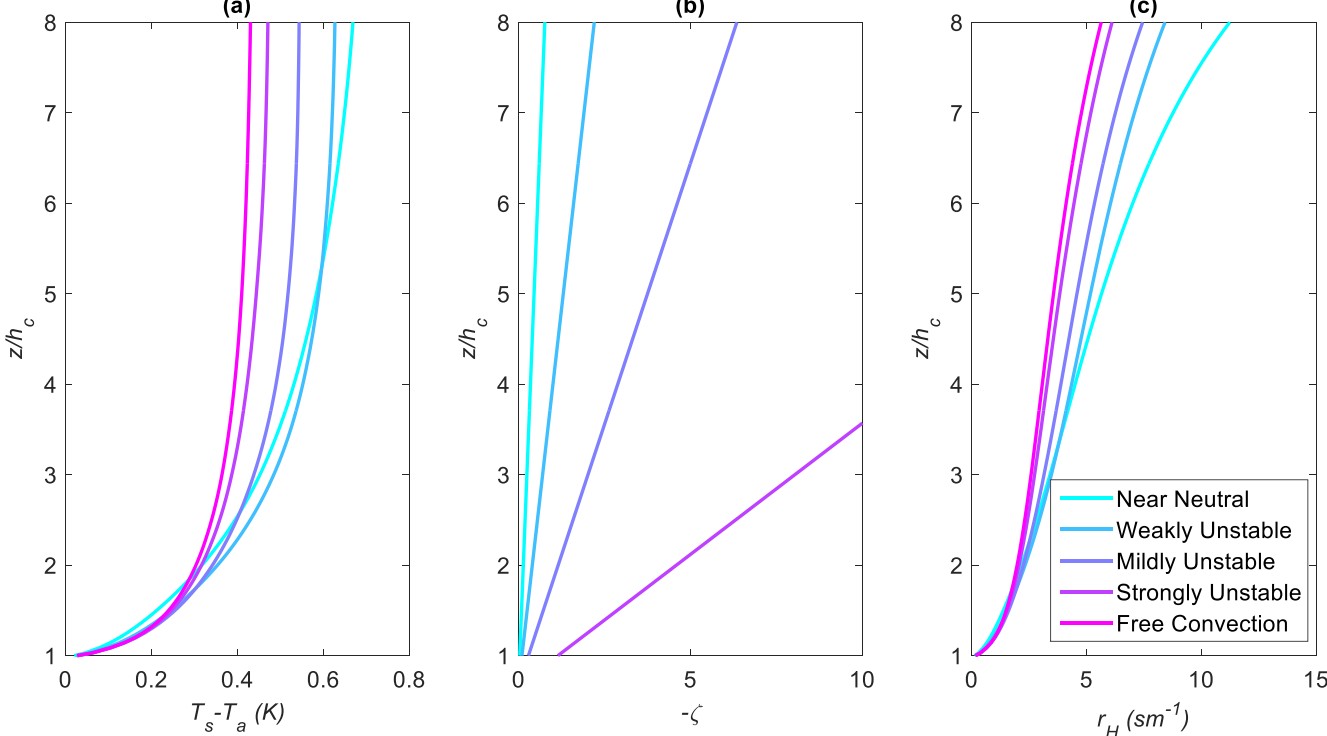

**Figure 2.** Aerodynamic resistance to heat transfer exhibiting canopy convector effect. Panel (a): Difference between surface and air temperature ($T_s - T_a$); panel (b): stability parameter $\zeta$; panel (c): canopy aerodynamic resistance ($r_H$).

instability, "convective updraughts developing at side walls of roughness elements extend upwards and provide extra resistances to the mean flow. Then the mean flow interacts with both solid obstacles and their virtual extensions (updraughts), which results in the increased roughness length". This increased roughness can be recognized as the aerodynamic roughness. For the same physical roughness of the canopy, increase of instability increases this aerodynamic roughness and in turn, reduces $r_H$. The low

5   aerodynamic resistance effectively allows larger eddies to form above the forest canopy which are more efficient to dissipate the sensible heat by promoting buoyancy. This description refers to a more general phenomenon as opposed to the the description by Rotenberg and Yakir (2011) which identifies the higher physical roughness of the canopy compared to the desert and is thus a more site specific description. Nevertheless, it is acknowledged that the more generic description presented here can be reconciled with the explanation from Rotenberg and Yakir (2011) by noting that increased physical roughness can also result

10   in increased aerodynamic roughness. Also incidentally, Rotenberg and Yakir (2010) reported a value of $r_H \approx 16$ for the Yatir forest which is of similar order of magnitude as what is found in panel (c) of figure 2. One important point to note in figure 1 is the magnitude of the Prandtl number, which is almost fixed to about 0.335 above the canopy. This can be reconciled with the theoretical prediction of the variation of $P_{r0}$ with stability by Li et al. (2015). For stability ranges $1 \leq -\zeta \leq 10$, $P_{r0}$ is also





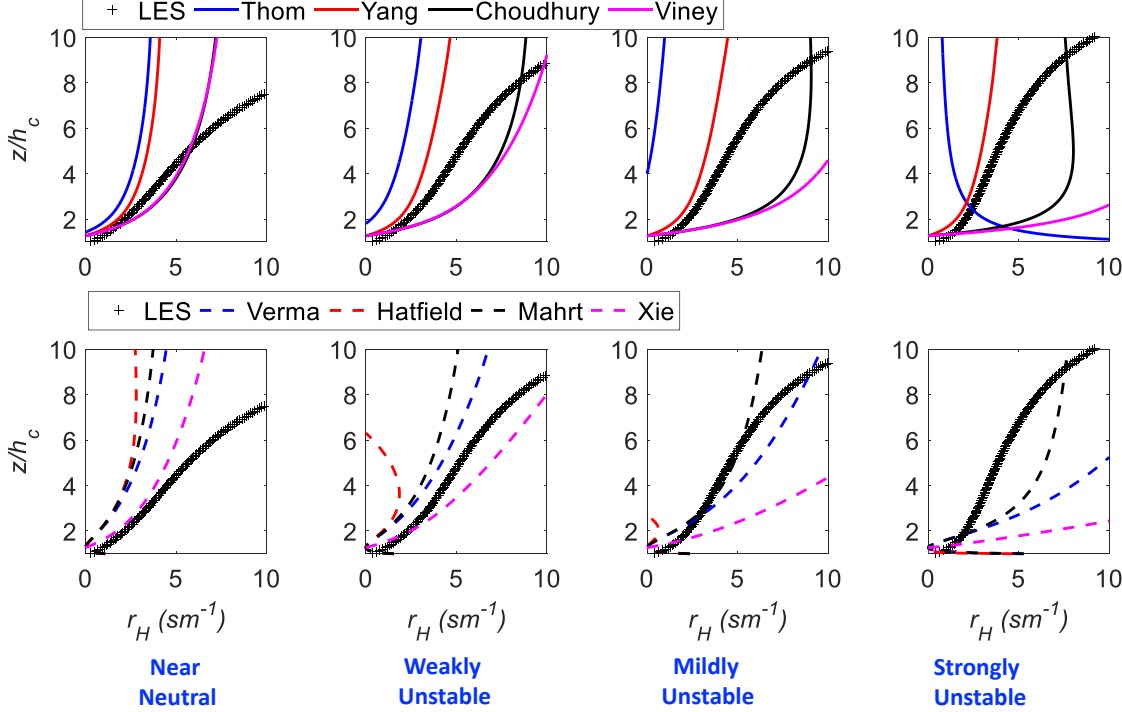

**Figure 3.** Variations of $r_H$ with height across stability ranges and comparisons with different parameterization schemes as described in table 1.

estimated to be approximately 0.33, consistent with the stability ranges plotted in figure 2. The variation of $P_{r0}$ with stability is discussed further in the appendix.

## 4.2 Testing different parameterizations

It is interesting to study if the different parameterizations capture the correct behavior of $r_H$ at different heights across stability. Figure 3 plots the variation of $r_H$ with height as obtained from the LES (black '+' markers), and the predicted $r_H$ from different parameterizations for the different stability cases - near neutral (column 1), weakly unstable (column 2), mildly unstable (column 3) and strongly unstable (column 4). The top row compares the parameterizations by Thom (1975) (blue line), Yang et al. (2001) (red line), Choudhury et al. (1986) (black line) and Viney (1991) (pink line) which assume $z_{0m} \neq z_{0h}$. The bottom panel compares the parameterizations by Verma et al. (1976) (blue dashed line), Hatfield et al. (1983) (red dashed line), Mahrt and Ek (1984) (black dashed line) and Xie (1988) (pink dashed line). It should be noted that a single value of roughness $z_{0m} = 0.6\,h_c$ has been chosen for all cases by trial and error to obtain a 'good' comparison in figure 3. As observed, none of the parameterizations can capture the correct height variations of $r_H$ except the one by Yang et al. (2001) for more unstable cases. However, all parameterizations seems to do a decent job close to the canopy top. This clearly indicates that one single





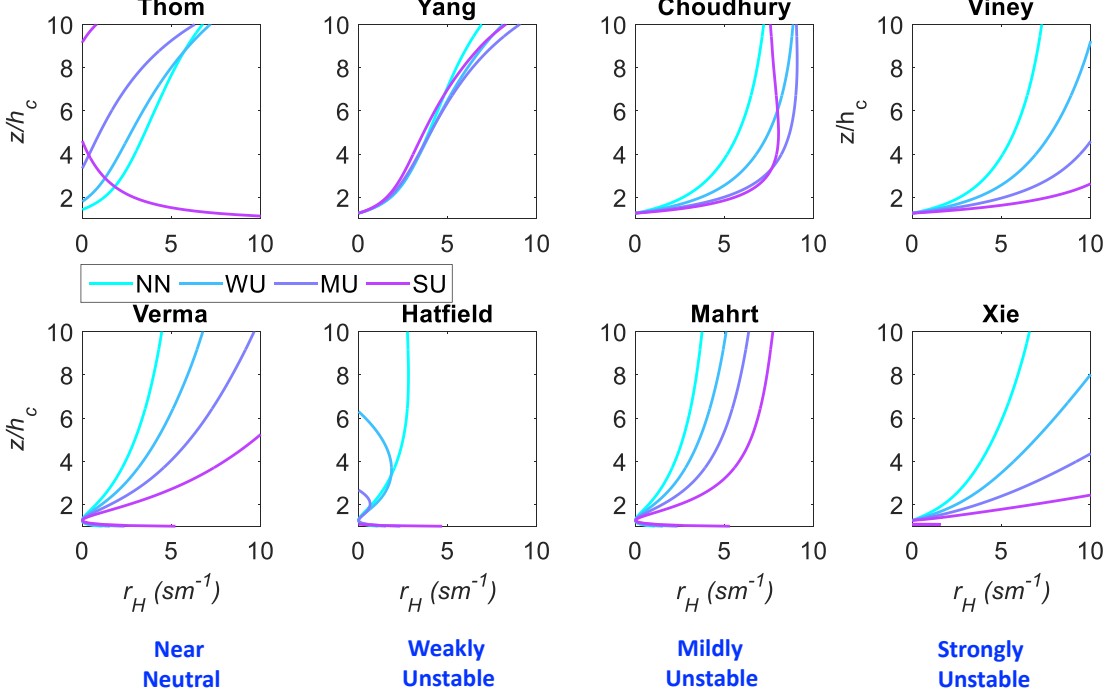

**Figure 4.** Variations of $r_H$ with height for different stability classes computed for each parameterization scheme as described in table 1.

value for $z_{0m}$ as suggested by these parameterizations is inadequate. To study if the different parameterization schemes can capture the canopy convector effect, $r_H$ computed from each method is plotted for different heights for the different stability classes in figure 4. The title of each panel describes which parameterization is plotted and the color shades starting from cyan to purple indicates increasing instability. As evident, only the parameterization by Thom (1975) captures the canopy convector

5   effect for weaker cases. The parameterization by Yang et al. (2001) also displays the signatures of CCE, however weakly. The other formulations cannot capture the correct trend of CCE at all. Thus at this stage, it is clear that Yang et al. (2001) formulation, based on MOST and distinguishing between two different roughness lengths is the most promising candidate to parametrize $r_H$ compared to the other formulations which apply some form of approximations or do not apply MOST.

### 4.3   Towards an improved parameterization for $r_H$

10  Until this stage, the momentum roughness length has been prescribed by trial and error and it warrants a more detailed investigation. To explore the effect of different roughness lengths, the parameterization by Yang et al. (2001) is computed across a wide range of $z_{0m}$ and compared with the LES outputs for the different stability classes. As observed, an increase of $z_{0m}$ with increasing instability captures the height variation better than a single roughness length for all stability classes further providing support for the notion put forward by Zilitinkevich et al. (2008). Hence the formulation by Yang et al. (2001) can be




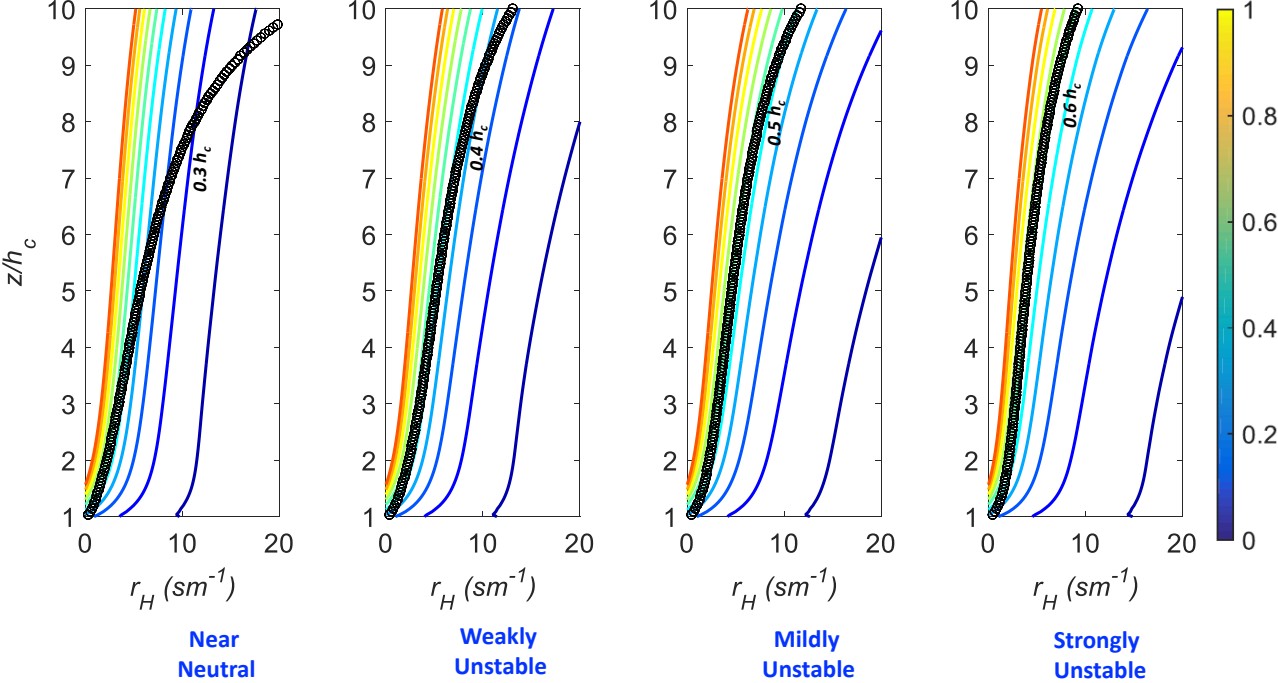

**Figure 5.** Variations of $r_H$ as given by the parameterization of Yang et al. (2001) with height across stability ranges and a wide range of $z_{0m}$. black '+' markers indicate the observed $r_H$ from LES at any particular stability state.

modified to include the effects of stratification on several parameters. Zilitinkevich et al. (2008) suggested a stability dependent zero-plane displacement length as well as a stability dependent $z_{0m}$ based on dimensional analysis, given by

$$d_s = \frac{d}{\left[1 + 0.56\left(\frac{h_c}{-L}\right)^{1/3}\right]}, \tag{11}$$

and

$$z_{0ms} = z_{0m}\left[1 + 1.15\left(\frac{h_c}{-L}\right)^{1/3}\right], \tag{12}$$

where $d_s$ and $z_{0ms}$ are the stability dependent zero-plane displacement length and roughness lengths for momentum respectively, $d$ and $z_{0m}$ being their neutral counterpart. $d = (2/3)h_c$ can be assumed as usual. The neutral $z_{0m}$ can be assumed to be related to LAI as given by Shuttleworth and Gurney (1990). According to the relation used by Shuttleworth and Gurney (1990), for an LAI of 5, a $z_{0m} = 0.12 h_c$ can be obtained (which is almost constant for a wide range of canopy drag coefficients and LAI). Moreover, if one uses the correct stability dependent Prandtl number $P_{r0}(\zeta)$ instead of setting it to unity, an improved





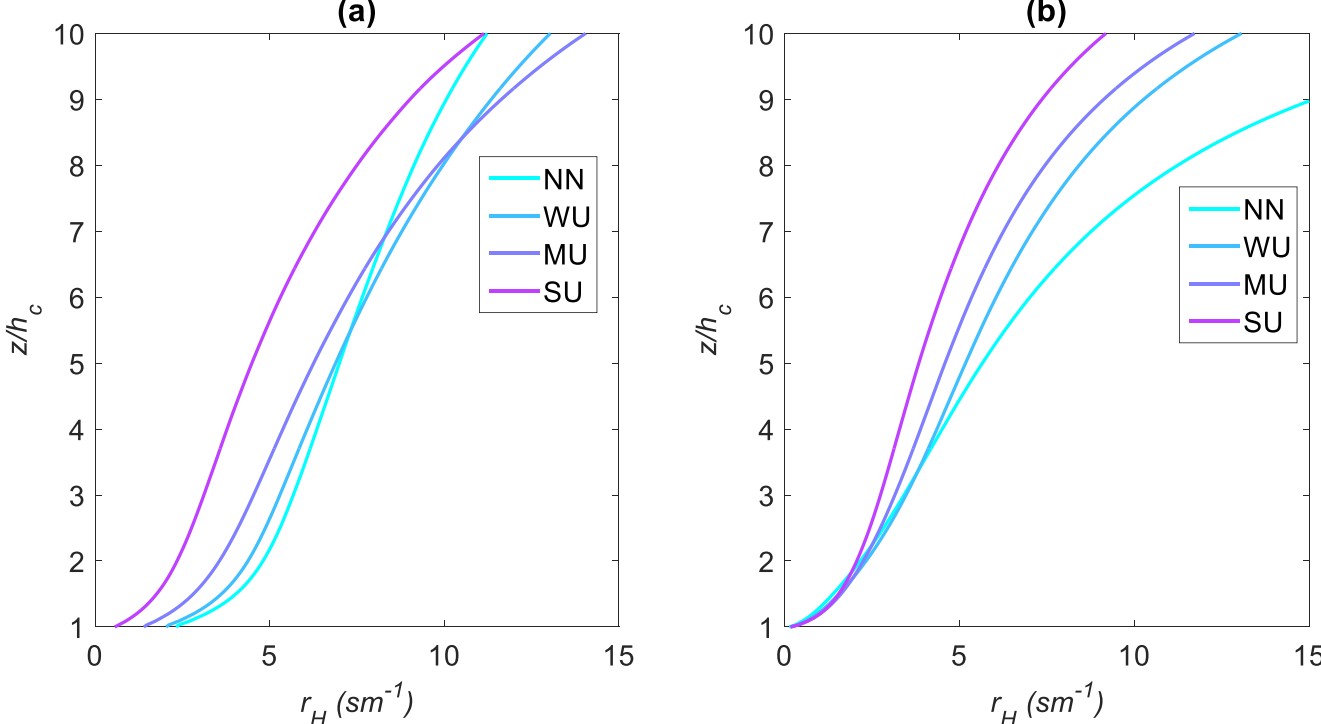

**Figure 6.** (a) The variation of $r_H$ with height across stability according to the improved formulation as given by equation 13;(b) similar variations of $r_H$ computed from the LES repeated again for comparison.

parameterization based on the Yang et al. (2001) can be written as

$$r_H = \frac{P_{r0}(\zeta)}{\kappa^2 u} \left[ \ln\left(\frac{z - d_s}{z_{0ms}}\right) - \psi_m\left(\zeta, \zeta_{0ms}\right) \right] \left[ \ln\left(\frac{z - d_s}{z_{0hs}}\right) - \psi_h\left(\zeta, \zeta_{0hs}\right) \right]. \tag{13}$$

Note that $z_{0ms}$ and $z_{0hs}$ are still related by the same relation $\kappa B^{-1} = \ln(z_{0ms}/z_{0hs})$ with $\kappa B^{-1} = 2.0$ as discussed earlier and $\zeta = (z - d_s)/L$ and $\zeta_{0ms} = z_{0ms}/L$, $\zeta_{0hs} = z_{0hs}/L$. If a Prandtl number of unity is still assumed but the roughness lengths are assumed to be varying with stability as given by equation 13 with a neutral value of $z_{0m} = 0.2\,h_c$, the formulation by Yang et al. (2001) is found to display the correct behavior of canopy convector effect with stability as observed in figure 6. Panel (a) shows the variation of $r_H$ with height across stability according to the improved formulation as given by equation 13. Panel (b) shows similar variations of $r_H$ computed from the LES repeated again for comparison. The profile for the near neutral case crosses over the more highly unstable cases at heights around $6\,h_c$, however, the general behavior of CCE is captured well. On the other hand, if the full complexity of equation 13 is used including a stability dependent Prandtl number (discussed in appendix A), but using the canopy top surface value of the sensible heat flux for all computations involved, the variation of modeled $r_H$ is shown on panel (a) of figure 7. $r_H$ computed from the LES results using the surface value of the heat flux is shown in panel (b). This assumption of a constant sensible heat flux in the canopy sub layer or the atmospheric surface layer





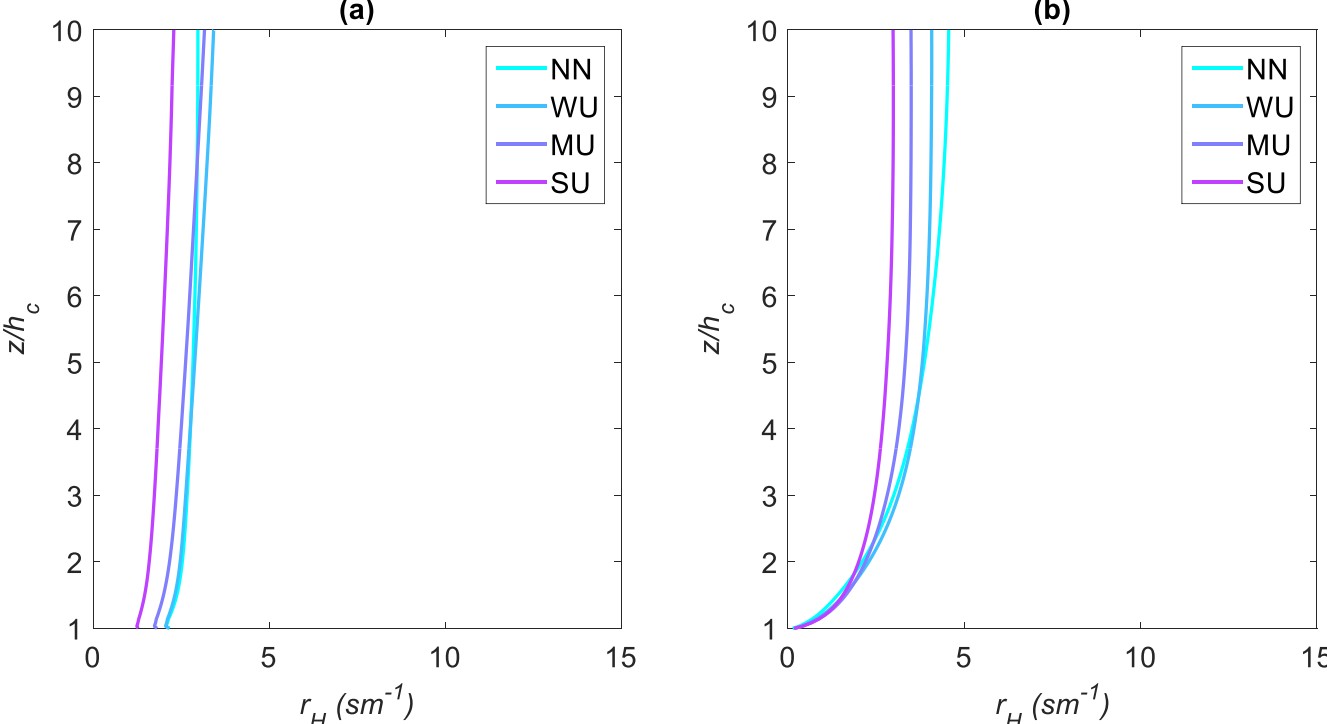

**Figure 7.** (a) The variation of $r_H$ with height across stability according to the improved formulation as modeled by equation 13, but using the top-of-canopy surface flux throughout all heights;(b) similar variations of $r_H$ computed from the LES using the top-of-canopy surface flux assumed to be constant in the surface layer.

is a more realistic one than a monotonically reducing sensible heat flux with height as shown in panel (e) of figure 1. In fact, the surface layer is defined as a constant flux layer (Stull, 2012). The reducing flux profiles in LES are common features of large-eddy simulations since the top boundary of the LES domain is assumed stress free (Shaw and Schumann, 1992). Figure 7 correctly captures the order of magnitude of the $r_H$ observed from the simulations, and also captures CCE correctly. However, it

5    is acknowledged that the exact profiles of the observed $r_H$ can not be captured. However, these different comparisons highlight the uncertainties involved in the parameterization of $r_H$.

## 5    Conclusion

The canopy aerodynamic resistance is a concept borrowed from the evapotranspiration literature where it represents the resistance between the idealized 'big-leaf' (a reduced order representation of the fully heterogeneous three dimensional canopy) and

10    the atmosphere for heat or vapor transfer (Alves et al., 1998). The Penman-Monteith equation to calculate evapotranspiration requires parameterization of the aerodynamic resistance which require information on roughness lengths for heat and momen-



tum and stability (Penman, 1948; Allen et al., 1998; Cleverly et al., 2013). $r_H$ parameterizations are also used in global climate models to describe the canopy-atmosphere interaction at the canopy surface layer (Walko et al., 2000). In semi arid ecosystems, vegetation canopies maintain a relatively cool surface temperature in spite of the high sensible heat flux by reducing the canopy aerodynamic resistance to heat transfer ($r_H$) — a phenomenon named 'canopy convector effect' by Rotenberg and

Yakir (2010). In the present work, a large-eddy simulation is used to examine this canopy convector effect and in the process, several existing parameterizations for $r_H$ is examined. The objectives behind this exploration is twofold. The first one is to investigate if the existing parameterizations exhibit canopy convector effect and the second one is to identify the uncertainties associated with these different parameterizations since they are applied in different climate models often under conditions of thermal stratification. As illustrated by the LES results, $r_H$ above the canopy are found to reduce systematically as the strength

of unstable stratification increases. This is deemed to be the core feature of canopy convector effect, since with increasing instability, more convective updraughts enhance the roughness over the canopy elements that the mean flow encounters. The height variation of $r_H$ is also found to have a highly nonlinear profile, thus any model prescribing a parameterization for $r_H$ needs to employ considerable caution regarding the height it is prescribed. Existing parameterizations of $r_H$ employ either Monin-Obukhov similarity theory (MOST) or Richardson number based empirical or semi-empirical formulations to account

for thermal stratification. However, most of them are found to be unable to describe the correct trend of CCE. Among different formulations, the one by Yang et al. (2001) is found to be the most promising candidate. This parameterization employs MOST, and accounts for stability parameters associated with roughness lengths for momentum and heat transfer. It is found out that a stability dependent zero-plane displacement height as well as stability dependent roughness lengths for momentum and heat transfer can improve its performance. Moreover, if the surface layer or the canopy sublayer is assumed to have a constant sen-

sible heat flux equal to the flux at the canopy top, and a stability dependent Prandtl number is used, the performance improves further. These assumption also leads to a less nonlinear height variation. These explorations highlight the uncertainties associated with the parameterizations of $r_H$. One possible major source of uncertainty is the usage of Monin-Obukhov similarity theory in the canopy sublayer (CSL) (up to $3h_c$ to $6h_c$) since it is not expected to perform in the CSL (Kaimal and Finnigan, 1994). Nevertheless, MOST formulations are found to outperform other semi-empirical formulations using Richardson

numbers. Thus future research work will involve studying these uncertainties of $r_H$ parameterizations in regional and global climate models. The consequence of this CCE on local circulation, atmospheric moisture and tree physiology will also be investigated, extending the preliminary study of Eder et al. (2015). However, the fact that CCE is a more generic feature of canopy turbulence provides hope that also the afforestation of an area larger than the Yatir forest would be able to cope with a high-radiation load under water scarcity in semi-arid climates.

*Acknowledgements.* This research was supported by the German Research Foundation (DFG) as part of the project "Climate feedbacks and benefits of semi-arid forests (CliFF)" and the project "Capturing all relevant scales of biosphere-atmosphere exchange - the enigmatic energy balance closure problem", which is funded by the Helmholtz-Association through the President's Initiative and Networking Fund, and by KIT. The authors thank the PALM group at Leibniz University Hannover for their open-source PALM code and also thank Dan Yakir and





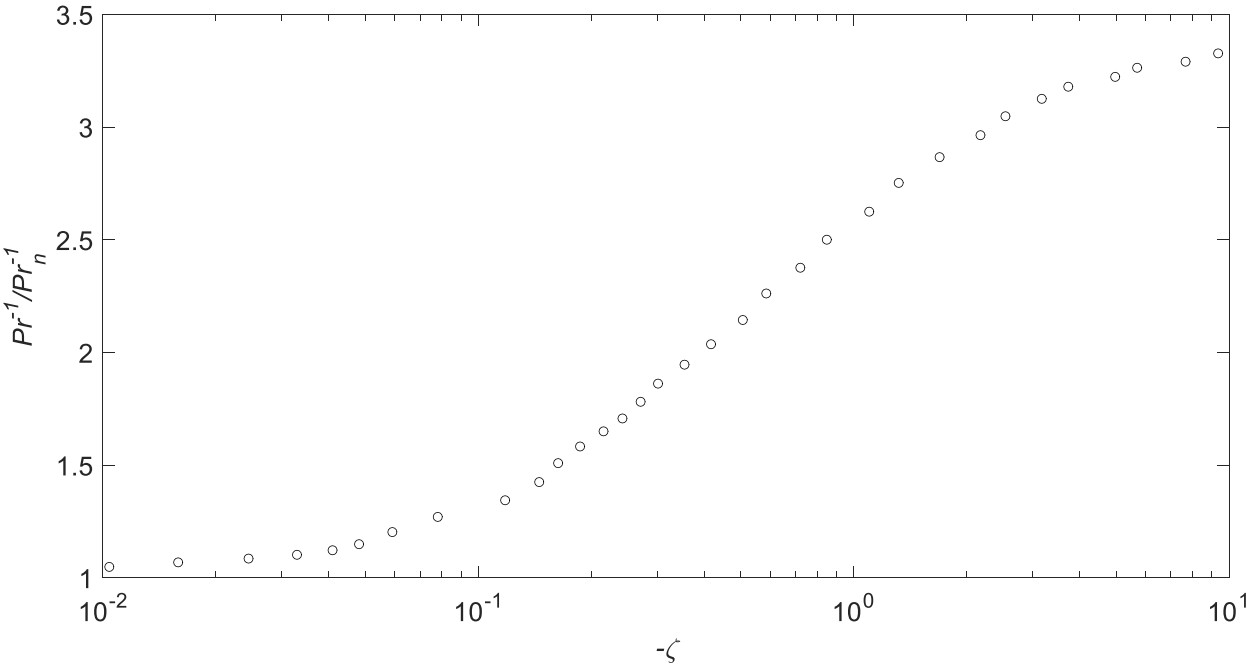

**Figure A1.** The variation of $Pr^{-1}/Pr_n^{-1}$ with stability according to the spectral budget formulation of Li et al. (2015).

Eyal Rotenberg at the Weizmann Institute of Science, Israel, for their support during the CliFF campaign. We also thank Prof. Thomas Foken, Bayreuth center of Ecology and environmental Research (BayCEER), University of Bayreuth, Germany, for his comments and suggestions.

## Appendix A: Appendix: Stability dependence of Prandtl number

The turbulent Prandtl number $P_{r0}$ is defined as the ratio of the eddy diffusivities of momentum and heat ($K_m/K_h$). The

5   variation of Prandtl number with stability ($P_{r0}(\zeta)$) was discussed in detail by Li et al. (2015) by using a spectral budget formulation and not repeated here. Only the predicted variation of $Pr^{-1}/Pr_n^{-1}$ with stability ($\zeta = z/L$) is digitized and produced on figure A1, which was experimentally validated by Li et al. (2015). $Pr_n^{-1}$ denotes the inverse of the neutral Prandtl number which can assumed to be equal to 1. Note that for the stability ranges computed in the LES simulations in figure 2, this formulation predicts a $P_{r0} \approx 0.33$, which is also observed in the $P_{r0}$ independently computed in figure 1.

10   *Author contributions.* T. Banerjee conceived the idea, conducted data analysis and wrote the paper. F. De Roo set up the Large Eddy Simulations. M. Mauder supervised the project and provided comments and suggestions.



*Competing interests.* The authors declare no conflict of interest.



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
