# Peer review of "Explaining the convector effect in canopy turbulence by means of large-eddy simulation"

_Hydrology and Earth System Sciences, 2017_

## Referee Comment (RC1) · G. Bohrer (Referee) · 23 Jan 2017

General Comments – the paper uses LES to determine the mechanisms responsible for increased heat flux from forests with lower albedo then their surroundings. This "convector effect" was previously described by observations. Nonetheless, this manuscript represents a very elegant approach to determine the theory behind this observed effect. They also provide a revised approach to parameterize this effect.

Specific comments: I would be happy if there was an explanation in simple terms of your proposed mechanism behind the convector effect. You are using very technical terms such as "atmospheric coupling", "aerodynamic resistance to heat transfer" and "aerodynamic roughness" but should spend a few lines in section 2.1 explaining this in layman terms, Will help to expand the readership of this, as it is relevant and interesting

for forest ecologists, managers and planers, and not only to forest meteorologists. If I got it correctly, the high leaf area of the forest reduces albedo, which leaves more of the incoming energy in the surface. However, the organization of these dark leaf surfaces is such that they are spread over a relatively thick canopy depth (relative to grassland of shrub land where all leaves are condensed in a much thinner layer). Because canopy in dry forests is sparse, wind can easily penetrate it (i.e., they have lower roughness length and displacement height) and can easily exchange heat with the leaf surfaces. Therefore, forests would have intrinsically lower aerodynamic resistance to heat transfer than shorter biomes (with the same leaf area) or other surfaces (with the same albedo). Given eq 1, that would mean higher heat flux.

Eq1 – by this equation, you are assuming the big-leaf equivalency. The reality is more complicated, as you identify later (P3 L5). I would restrict the statement in P2 line 26 "it is important to recall that, when adopting the simplified big-leaf representation of the forest as a single surface". Furthermore, I would call $r\_H$ the "apparent canopy aerodynamic resistance to heat transfer" (line 29) to indicate that this property is a construct of the formulation and not a direct physical property.

The entire roughness length formulation (eq 2-8 and table 1) is based on different variants of analytic approximation approaches to reduce the complexity of flow in and above the forest canopy to a 2-D-surface equivalent. It is widely accepted that as you go near the canopy, the MOST approach is not valid. It was proposed (and rather widely accepted) that a mixing length-driven approach can be applied (see Harman and Finnigan 2007, 2008 BLM). Nonetheless, large-scale models, which cannot vertically resolve the canopies, still use MOST and it has been demonstrated to be relatively accurate. My point here is that this section of the manuscript (eq1-8) should not be mixed with the notion that it explains the physics, but state very clearly that it revisits the current leading approach for simplification of the physics in a parameterized way that can be used by coarse-resolution models. I think it will be beneficial for the manuscript if you emphasize the point that you are using an LES with an explicit 3-D canopy, where

the surface assumptions are not needed to develop a revised approximation approach for the surface-equivalence that account to the forest density effects and parameterize for its outcomes in a way that will allow resolving the heat convector effect even in large-scale models.

P3 L26 "d is the zero-plane displacement height taken as 2/3h_c..." will be more correct to say: d is the zero-plane displacement height, often approximated as 2/3h_c... (and see study of roughness length and displacement height in a forest stand and their best approximation – Maurer et al Biogeosciences, 12, 2533–2548, 2015, and Maurer et al Agricultural and Forest Meteorology 177 (2013) 24– 34)

Technical corrections: I recommend setting an acronym for Rotenberg and Yakir (2010) - RY10, after the first use. It is listed so many ties that it gets rather tedious.

Pg 2, L 32: "This 'canopy convector effect' is sufficiently efficient... Word placement is confusing. Can change "sufficiently efficient" to 'adequate enough', 'suitably efficient'

Pg 6, L 26: "while the output of first 6400 s..." change to "while the output of the first 6400 s"

Table 2 Please include the meaning of the Stability class (e.g. "Near-Neutral", Weakly Unstable"...) as the first column of the table. Will make it easier to remember, and look up.

Pg 9, L 6: "This description refers to a more general phenomenon as opposed to the the description...". Remove "the"

Pg 15, L21: Change to "These assumptions also lead to a less nonlinear height variation".

---

## Referee Comment (RC2) · Anonymous Referee #2 · 8 Feb 2017

**General comments:**

This manuscript is motivated by the work from Rotenberg and Yakir (2010), which observed a decrease in aerodynamic resistance to heat transfer ($r_H$) over a forest when compared to a shrubland region under similar conditions, an effect that is caused by an increase in surface roughness and is accompanied by an increase in atmospheric instability. This effect was called "canopy convector effect" (CCE) by Rotenberg and Yakir (2010). In this manuscript, the authors investigate the occurrence of CCE above the canopy using Large Eddy Simulation (LES). After observing a decrease in $r_H$ with increase in unstable conditions in the simulations (used as evidence of CCE), the authors compare different models of $r_H$ as a function of height (above the canopy) and atmospheric stability with the simulation results, and conclude that some models cannot capture the correct trend of CCE at all (because they present an increase in $r_H$ with instability), and only two (out of eight) models display the signature of CCE (decrease in $r_H$ with instability). The authors proposed an improved parameterization of one of the $r_H$ models by using a value of momentum roughness length scale $z_{0m}$ that vary with atmospheric stability, improving the agreement between model and LES results. The authors conclude that CCE is a generic feature of canopy turbulence.

Because the value of $r_H$ is needed for a wide range of applications, the investigation of the behavior of different $r_H$ parameterizations above the canopy is useful, and the use of LES for this purpose is appropriate, therefore this manuscript deals with an interesting topic. However, as described in more details below, I believe the manuscript needs an alternative motivation, better description of the simulations and models and better interpretation of the results.

**Major specific comments:**

1. I'm not sure I agree with the authors' interpretation of what CCE represents. The authors defined CCE as a decrease in aerodynamic resistance above the canopy, which can be accomplished by an increase in atmospheric instability. In my opinion, it is already well-accepted that there is an increase in turbulent transport (estimated by eddy diffusivity parameters, for example) and consequent decrease in $r_H$ with increasing instability. This should be valid over a canopy and over bare soil. The differences between the canopy and the bare soil cases are the type of the turbulent flow and level of penetration of the transporting eddies across the heat source layer in the canopy case (compared to the no-penetration condition over bare soil), which makes the turbulent transport different in the canopy case compared to the bare soil case, even if all other factors are the same. In Equation (1) this difference is accounted by reducing the aerodynamic resistance in the canopy case, and I think this is what Rotenberg and Yakir (2010) meant in the

definition of CCE. Therefore, a study that wants to better describe the CCE phenomenon should focus on comparing turbulent transport characteristics across different canopies and bare soil, probably for different stabilities, but not only the stability difference in one canopy, as this difference is already expected. Therefore, I believe that the CCE should not be the motivation of this manuscript.

2. I'm surprised with the results of increased $r_H$ with increase negative $Ri_B$ (increase instability) for the non-MOST models. I believe all models try to replicate the overall idea that turbulent transport increases with instability, and after a quick look on the equations and original manuscripts, it seems to me that $r_H$ should decrease with instability in all models, therefore I'm confused about the results shown in Figures 3 and 4.

3. The description of the temperature field simulated with LES is not complete. Dias-Junior et al. (2015) simulated only a near neutral case, and Patton et al. (2016) included a source profile term in the temperature equation which comes from the land-surface model, which is not present in the simulations presented here, therefore they cannot be used as references for some of the details of the simulations performed here. In Table 2, the value of $\overline{w'T'}_s$ is defined as being at the ground, where the same values used in Patton et al. (2016) were defined at canopy top. When looking into Figure 1, it is not clear where the imposed heat flux value is, as nowhere in the profile there is a match with the imposed values. If the heat source is applied in the SGS part of the model, and Figure 1(e) shows only the resolved part, maybe the resolved + SGS part of the heat flux should be presented instead. Also, the final profiles of temperature have a peak at canopy top, also different from the results obtained by Patton et al. (2016). Although this may not affect the final conclusions, the equations, sources and boundary conditions used in the temperature field of LES need to be clarified.

**Minor specific comments:**

- Introduction: include a paragraph describing why better estimations of $r_H$ are needed, even though this is a parameter poorly defined for atmospheric transport. I believe the conclusion has some of the information that could be in the intro.

- Section 2.2: emphasize here that these models were developed for conditions different from canopy sublayer, again this is in the conclusion but should be discussed earlier in the manuscript. This can be a major cause of discrepancies between the models and the simulation, which could be tested by performing simulations without canopy and comparing with the models. After a quick look, I could not find such a test in the literature.

- Section 4.1: if possible, when describing the figures in the text, give some justification of the result encountered, for example, if the variations with instability observed makes physical sense.

- Page 7, line 21: mention how the eddy diffusivities were estimated.

- Page 10, line 6: describe how these profiles where estimated. Which values came from LES, which are constant, which are a function of height, for example?

- Page 11, line 5: not clear what "for weaker cases" mean. Do you mean for weaker instabilities?

**Technical corrections and minor suggestions:**

- Why "(in)stability"?
- Figure 3: I suggest to use $z/z_i$ (where $z_i$ is the top of the ABL) instead of $z/h$, emphasizing that the entire plot is above the canopy. It can help to discuss the region where MOST (and therefore some of the models) is valid (surface layer).

- Figure 4: I believe that the blue captions below the figures are wrong.

**References**

Dias-Junior, C. Q., Marques Filho, E. P., and Sá, L. D. A. (2015). A large eddy simulation model applied to analyze the turbulent flow above amazon forest. *Journal of Wind Engineering and Industrial Aerodynamics*, 147:143 – 153.

Patton, E. G., Sullivan, P. P., Shaw, R. H., Finnigan, J. J., and Weil, J. C. (2016). Atmospheric stability influences on coupled boundary layer and canopy turbulence. *Journal of the Atmospheric Sciences*, 73(4):1621–1647.

Rotenberg, E. and Yakir, D. (2010). Contribution of semi-arid forests to the climate system. 327(5964):451–454.

---

## Author Comment (AC1) · 4 Apr 2017

Response to reviewer 1 (Prof. Gil Bohrer, Ohio State University)

**General Comments – the paper uses LES to determine the mechanisms responsible for increased heat flux from forests with lower albedo then their surroundings. This "convector effect" was previously described by observations. Nonetheless, this manuscript represents a very elegant approach to determine the theory behind this observed effect. They also provide a revised approach to parameterize this effect.**

*We sincerely thank Prof. Bohrer for the generous comments and the constructive suggestions.*

**Specific comments: I would be happy if there was an explanation in simple terms of your proposed mechanism behind the convector effect. You are using very technical terms such as "atmospheric coupling", "aerodynamic resistance to heat transfer" and "aerodynamic roughness" but should spend a few lines in section 2.1 explaining this in layman terms, Will help to expand the readership of this, as it is relevant and interesting for forest ecologists, managers and planers, and not only to forest meteorologists. If I got it correctly, the high leaf area of the forest reduces albedo, which leaves more of the incoming energy in the surface. However, the organization of these dark leaf surfaces is such that they are spread over a relatively thick canopy depth (relative to grassland of shrub land where all leaves are condensed in a much thinner layer). Because canopy in dry forests is sparse, wind can easily penetrate it (i.e., they have lower roughness length and displacement height) and can easily exchange heat with the leaf surfaces. Therefore, forests would have intrinsically lower aerodynamic resistance to heat transfer than shorter biomes (with the same leaf area) or other surfaces (with the same albedo). Given eq 1, that would mean higher heat flux.**

*We thank Prof. Bohrer for pointing this out. This a valuable suggestion. We have added the following text in section 2.1:*

"Therefore to summarize canopy convector effect in simpler terms it can be mentioned that the darker and colder canopy surface reduces albedo, which leaves more of the incoming energy on the canopy surface. However, the organization of these dark leaf surfaces is such that they are spread over a relatively thick canopy depth (relative to grassland or shrub land where all leaves are condensed in a much thinner layer). Because canopy in dry forests is sparse, wind can easily penetrate it and can easily exchange heat with the leaf surfaces. Therefore, forests would have intrinsically lower aerodynamic resistance to heat transfer than shorter biomes because of the higher roughness.

Moreover, the same forest (with the same *physical roughness*) could have higher *aerodynamic roughness* and consequently lower aerodynamic resistance to heat transfer for more heat stressed conditions. Given eq 1, that would mean higher heat flux. Thus while CCE would always be present in a forest compared to a grass or shrubland because of the obvious roughness difference, we establish that CCE can also be present within the same forest for different conditions of heat stress- which is a more subtle point and will be further discussed in the following sections by using large eddy simulations (LES)."

**Eq1 – by this equation, you are assuming the big-leaf equivalency. The reality is more complicated, as you identify later (P3 L5). I would restrict the statement in P2 line 26 "it is important to recall that, when adopting the simplified big-leaf representation of the forest as a single surface".**

*Agreed and added this sentence:* "it is important to recall that, when adopting the simplified big-leaf representation of the forest as a single surface"

**Furthermore, I would call r_H the "apparent canopy aerodynamic resistance to heat transfer" (line 29) to indicate that this property is a construct of the formulation and not a direct physical property.**

*Agreed and modified accordingly. Also added the text:* "the word apparent is used to indicate that this property is a construct of the formulation and not a direct physical property"

**The entire roughness length formulation (eq 2-8 and table 1) is based on different variants of analytic approximation approaches to reduce the complexity of flow in and above the forest canopy to a 2-D-surface equivalent. It is widely accepted that as you go near the canopy, the MOST approach is not valid. It was proposed (and rather widely accepted) that a mixing length-driven approach can be applied (see Harman and Finnigan 2007, 2008 BLM). Nonetheless, large-scale models, which cannot vertically resolve the canopies, still use MOST and it has been demonstrated to be relatively accurate. My point here is that this section of the manuscript (eq1-8) should not be mixed with the notion that it explains the physics, but state very clearly that it revisits the current leading approach for simplification of the physics in a parameterized way that can be used by coarse-resolution models**

*Agreed. The following text is added to the end of the section 2.2:*

"Before moving on to the usage of LES, it warrants mentioning that the entire roughness length formulation (equation 2 - 8 and table 1) is based on different variants of analytic

approximation approaches to reduce the complexity of flow in and above the forest canopy to a 2-D-surface equivalent. It is widely accepted that the MOST approach is not completely accurate close to the canopy (Foken, 2006). It was proposed that a mixing length-driven approach can be applied (Harman and Finnigan, 2007). Nonetheless, large-scale models, which cannot vertically resolve the canopies, still use MOST and it has been demonstrated to be relatively accurate. Thus from an operational perspective, the present formulation revisits the current leading approach for simplification of the physics in a parameterized way that can be used by coarse-resolution models."

**I think it will be beneficial for the manuscript if you emphasize the point that you are using an LES with an explicit 3-D canopy, where the surface assumptions are not needed to develop a revised approximation approach for the surface-equivalence that account to the forest density effects and parameterize for its outcomes in a way that will allow resolving the heat convector effect even in large-scale models.**

*Agreed and the following text is added in section 3*: "It is worth highlighting again here that the large eddy simulations have been conducted with an explicit 3-D canopy. This means that the surface assumptions are not needed to develop a revised approximation approach for the surface-equivalence that accounts for the forest density effects. Only the outcomes of the LES are parameterized in a way that will allow resolving the canopy convector effect even in large-scale models".

**P3 L26 "d is the zero-plane displacement height taken as 2/3h_c. . ." will be more correct to say: d is the zero-plane displacement height, often approximated as 2/3h_c. . . (and see study of roughness length and displacement height in a forest stand and their best approximation – Maurer et al Biogeosciences, 12, 2533–2548, 2015, and Maurer et al Agricultural and Forest Meteorology 177 (2013) 24– 34)**

*Agreed and modified accordingly. The references are added as well.*

**Technical corrections: I recommend setting an acronym for Rotenberg and Yakir (2010) - RY10, after the first use. It is listed so many ties that it gets rather tedious.**

*Agreed and changed accordingly.*

**Pg 2, L 32: "This 'canopy convector effect' is sufficiently efficient. . . Word placement is confusing. Can change "sufficiently efficient" to 'adequate enough', 'suitably efficient'**

*Agreed and changed to 'adequate enough'.*

**Pg 6, L 26: "while the output of first 6400 s. . ." change to "while the output of the first 6400 s"**

*Changed to* "while the output of the first 6400 s"

**Table 2 Please include the meaning of the Stability class (e.g. "Near-Neutral", Weakly Unstable". . .) as the first column of the table. Will make it easier to remember, and look up.**

*Added the meanings of the stability classes to the table.*

**Pg 9, L 6: "This description refers to a more general phenomenon as opposed to the the description. . .". Remove "the"**

*Removed the extra "the"*

**Pg 15, L21: Change to "These assumptions also lead to a less nonlinear height variation".**

*Changed to* "These assumptions also lead to a less nonlinear height variation".

---

## Author Comment (AC2) · 4 Apr 2017

**Response to reviewer 2.**

**This manuscript is motivated by the work from Rotenberg and Yakir (2010), which observed a decrease in aerodynamic resistance to heat transfer ($r_H$) over a forest when compared to a shrubland region under similar conditions, an effect that is caused by an increase in surface roughness and is accompanied by an increase in atmospheric instability. This effect was called "canopy convector effect" (CCE) by Rotenberg and Yakir (2010). In this manuscript, the authors investigate the occurrence of CCE above the canopy using Large Eddy Simulation (LES). After observing a decrease in $r_H$ with increase in unstable conditions in the simulations (used as evidence of CCE), the authors compare different models of $r_H$ as a function of height (above the canopy) and atmospheric stability with the simulation results, and conclude that some models can not capture the correct trend of CCE at all (because they present an increase in $r_H$ with instability), and only two (out of eight) models display the signature of CCE (decrease in $r_H$ with instability). The authors proposed an improved parameterization of one of the $r_H$ models by using a value of momentum roughness length scale $z_{0m}$ that vary with atmospheric stability, improving the agreement between model and LES results. The authors conclude that CCE is a generic feature of canopy turbulence. Because the value of $r_H$ is needed for a wide range of applications, the investigation of the behavior of different $r_H$ parameterizations above the canopy is useful, and the use of LES for this purpose is appropriate, therefore this manuscript deals with an interesting topic. However, as described in more details below, I believe the manuscript needs an alternative motivation, better description of the simulations and models and better interpretation of the results.**

*We thank the reviewer for the nice summary and the constructive comments. We have attempted to respond to all the points raised by the reviewer.*

**Major specific comments:**

**I'm not sure I agree with the authors' interpretation of what CCE represents. The authors defined CCE as a decrease in aerodynamic resistance above the canopy, which can be accomplished by an increase in atmospheric instability. In my opinion, it is already well-accepted that there is an increase in turbulent transport (estimated by eddy diffusivity parameters, for example) and consequent decrease in $r_H$ with increasing instability. This should be valid over a canopy and over bare soil. The differences between the canopy and the bare soil cases are the type of the turbulent flow and level of penetration of the transporting eddies across the heat source layer in the canopy case (compared to the no-penetration condition over bare soil), which makes the**

**turbulent transport different in the canopy case compared to the bare soil case, even if all other factors are the same. In Equation (1) this difference is accounted by reducing the aerodynamic resistance in the canopy case, and I think this is what Rotenberg and Yakir (2010) meant in the definition of CCE. Therefore, a study that wants to better describe the CCE phenomenon should focus on comparing turbulent transport characteristics across different canopies and bare soil, probably for different stabilities, but not only the stability difference in one canopy, as this difference is already expected. There- fore, I believe that the CCE should not be the motivation of this manuscript**

*It is agreed that the earlier version of the manuscript was not very clear on this issue. We have added the following text to make it more clear:*

""Therefore to summarize canopy convector effect in simpler terms it can be mentioned that the darker and colder canopy surface reduces albedo, which leaves more of the incoming energy on the canopy surface. However, the organization of these dark leaf surfaces is such that they are spread over a relatively thick canopy depth (relative to grassland or shrub land where all leaves are condensed in a much thinner layer). Because canopy in dry forests is sparse, wind can easily penetrate it and can easily exchange heat with the leaf surfaces. Therefore, forests would have intrinsically lower aerodynamic resistance to heat transfer than shorter biomes because of the higher roughness. Moreover, the same forest (with the same physical roughness) could have higher aerodynamic roughness and consequently lower aerodynamic resistance to heat transfer for more heat stressed conditions. Given eq 1, that would mean higher heat flux. Thus while CCE would always be present in a forest compared to a grass or shrubland because of the obvious roughness difference, we establish that CCE can also be present within the same forest for different conditions of heat stress- which is a more subtle point and will be further discussed in the following sections by using large eddy simulations (LES)."

**I'm surprised with the results of increased $r_H$ with increase negative $Ri_B$ (in- crease instability) for the non-MOST models. I believe all models try to replicate the overall idea that turbulent transport increases with instability, and after a quick look on the equations and original manuscripts, it seems to me that $r_H$ should decrease with instability in all models, therefore I'm confused about the results shown in Figures 3 and 4.**

*This is a good point. The $Ri_B$ at each level has been calculated by the following equation (eq 9 in the paper)*

$$Ri_B = \frac{g}{T_a} \frac{(T_a - T_s)(z - d)}{U_{\parallel}^2}$$

*where the LES profiles have been used at each level. Thus $Ri_B$ has typical profiles for two instability cases shown in the following figures (1) and (2). As observed, the negative $Ri_B$ at one particular height is higher for the strongly unstable case compared to the weakly unstable case. However, the empirical formulations are highly nonlinear and the effects of the height variations of the profiles of the other parameters (such as U) are manifested in the results in figure 3 and 4. This result is not obvious and that is what motivated our study partly.*

[Figure]

*If all other parameters are fixed, we would see a variation of rH reducing with increasing negative $Ri_B$ (as You point out correctly) as shown in the following figure (3) which used the formulation of Xie (1998) as an example. Here all parameters are fixed and only RiB is varied. So while Your suggestion is correct, there is no conflict with the results in the paper. The formulations are used correctly (we checked them again) as they can reproduce the correct variations as suggested in the corresponding references if all other parameters are*

*held constant- it is the nonlinear height variations of all other profiles that creates the reported variations in this paper.*

[Figure]

(figure 3)

**The description of the temperature field simulated with LES is not complete. Dias-Junior et al. (2015) simulated only a near neutral case, and Patton et al. (2016) included a source profile term in the temperature equation which comes from the land-surface model, which is not present in the simulations presented here, there- fore they cannot be used as references for some of the details of the simulations performed here. In Table 2, the value of $\overline{w'T'}_s$ is defined as being at the ground, where the same values used in Patton et al. (2016) were defined at canopy top. When looking into Figure 1, it is not clear where the imposed heat flux value is, as nowhere in the profile there is a match with the imposed values. If the heat source is applied in the SGS part of the model, and Figure 1(e) shows only the resolved part, maybe the resolved + SGS part of the heat flux should be presented instead. Also, the final profiles of temperature have a peak at canopy top, also different from the results obtained by Patton et al. (2016). Although this may not affect the final conclusions, the equations, sources and boundary conditions used in the temperature field of LES need to be clarified**

*Although the final conclusions are not affected, we admit that we have been unclear here and thank the reviewer for pointing this out. We have a heat flux imposed at the top of the canopy*

*and within the canopy there is an exponential decay of the incoming energy due to the absorption and reflection by the leaves. The prescribed heat flux value at the ground surface only becomes effective for grid points that do not have a canopy layer above. Therefore for gridpoints with canopy our setup indeed differs from Patton et al due to the strong absorption of solar radiation within the canopy, which also explains the peak in the temperature profile at the top of the canopy in our case. The reviewer is also correct that Fig 1e) presents only the resolved part of the flux. We have corrected this and with the subgrid part added, the output value at of the top of the canopy equals the value imposed in the simulation.*

*To quote the PALM manual:*
*"The heat source distribution is calculated by a decaying exponential function of the downward cumulative leaf area index (integral of the leaf area density), assuming that the foliage inside the plant canopy is heated by solar radiation penetrating the canopy layers according to the distribution of net radiation as suggested by Brown & Covey (1966; Agric. Meteorol. 3, 73-96)).This approach has been applied e.g. by Shaw & Schumann (1992; Bound.-Layer Meteorol. 61, 47-64)."*

*We have clarified this in the text and also added the necessary equations and boundary conditions in the appendix:*

**B1   surface heat flux formulation and boundary conditions**

The ground surface heat flux for gridpoints with a canopy layer is given by:

$$\overline{w'T'}_s = \overline{w'T'}_{toc} \times \exp\left(-\epsilon_c \int_0^{h_c} LAD(z)\mathrm{d}z\right), \tag{B1}$$

with $\epsilon_c = 0.6$ the extinction coefficient of light within the canopy. Within the canopy the plant-canopy heating rate is calculated as the vertical divergence of the canopy heat fluxes:

$$\overline{w'T'}_{toc} \frac{d}{dz} \exp\left(\epsilon_c \int_z^{h_c} LAD(z')\mathrm{d}z'\right) \tag{B2}$$

The bottom boundary condition for potential temperature is a Neumann condition, the boundary condition at the top of the domain is such that the initial temperature gradient is maintained at the top of the domain.

**Minor specific comments:**

**Introduction: include a paragraph describing why better estimations of $r_H$ are needed, even though this is a parameter poorly defined for atmospheric transport. I believe the conclusion has some of the information that could be in the intro.**

*Agreed. The following text has been added:*

"The canopy aerodynamic resistance is a concept borrowed from the evapotranspiration literature where it represents the resistance between the idealized 'big-leaf' (a reduced order representation of the fully heterogeneous three dimensional canopy) and the atmosphere for heat or vapor transfer (Monteith, 1973; Foken et al., 1995; Alves et al., 1998; Monteith and Unsworth, 2007). The Penman-Monteith equation to calculate evapotranspiration requires parameterization of the aerodynamic resistance which require information on roughness lengths for heat and momentum and stability (Penman, 1948; Allen et al., 1998; Cleverly et al., 2013). $r_H$ parameterizations are also used in global climate models to describe the canopy-atmosphere interaction at the canopy surface layer (Walko et al., 2000). Thus better parameterizations of $r_H$ are of fundamental importance in modeling canopy level fluxes of heat and water vapor which can be used in assessing impacts of climate change, disturbance effects such as vegetation thinning, forest fires etc., as well as for developing forest management strategies."

**Section 2.2: emphasize here that these models were developed for conditions different from canopy sublayer, again this is in the conclusion but should be discussed earlier in the manuscript. This can be a major cause of discrepancies between the models and the simulation, which could be tested by performing simulations without canopy and comparing with the models. After a quick look, I could not find such a test in the literature.**

*This is a valid point and also has been pointed out by the other reviewer. The following text has been added to section 2.2:*

"Before moving on to the usage of LES, it warrants mentioning that the entire roughness length formulation (equation 2 - 8 and table 1) is based on different variants of analytic approximation approaches to reduce the complexity of flow in and above the forest canopy to

a 2-D-surface equivalent. It is widely accepted that the MOST approach is not completely accurate close to the canopy (Foken, 2006). It was proposed that a mixing length-driven approach can be applied (Harman and Finnigan, 2007). Nonetheless, large-scale models, which cannot vertically resolve the canopies, still use MOST and it has been demonstrated to be relatively accurate. Thus from an operational perspective, the present formulation revisits the current leading approach for simplification of the physics in a parameterized way that can be used by coarse-resolution models."

**Section 4.1: if possible, when describing the figures in the text, give some justification of the result encountered, for example, if the variations with instability observed makes physical sense.**

*The following text has been added.*

"These results are physical consistent. The near neutral case is dominated by mechanical shear driven turbulence - given by the highest mean velocity. The free convection case is fully buoyancy driven and the motion is fully upwards-as evident by the near zero mean horizontal velocity. For the same reasons, the turbulent intensity and friction velocity follow the same pattern. The strongly unstable cases have highest heat fluxes, which is also physically consistent. "

**Page 7, line 21: mention how the eddy diffusivities were estimated.**

*We have added this in an appendix:*

**B2 Eddy diffusivity formulation**

The computation for the eddy diffusivities in PALM follows the standard procedure for 1.5 order turbulence closure. Thus they are computed from the subgrid-scale turbulent kinetic energy, more precisely equations (13-14) from Maronga et al. (2015).

Eddy diffusivity for momentum:

$$K_m = c_m l \sqrt{e} \tag{B3}$$

Eddy diffusivity for heat:

$$K_h = \left(1 + \frac{2l}{\Delta}\right) K_m \tag{B4}$$

With $e$ the subgrid-scale turbulent kinetic energy (a prognostic variable), $c_m = 0.1$, and $\Delta$ the geometric mean of the grid spacings in x,y and z. Finally, $l$ is the subgrid-scale mixing length depending on $\Delta$, stability, and distance from the topography elements or ground surface.

**Page 10, line 6: describe how these profiles where estimated. Which values came from LES, which are constant, which are a function of height, for example?**

*The following text has been added*: "To compute $r_H$ variations, the LES generated profiles of mean velocity u, sensible heat flux, air temperature, Prandtl number (thus the diffusivities) are used where all of them have z variations. The friction velocity $u_*$ and the roughness lengths are fixed."

**Page 11, line 5: not clear what "for weaker cases" mean. Do you mean for weaker instabilities?**

*Changed to weaker instabilities.*

**Technical corrections and minor suggestions:**
**• Why "(in)stability"?**

*Removed the brackets.*

**Figure 3: I suggest to use $z/z_i$ (where $z_i$ is the top of the ABL) instead of z/h, emphasizing that the entire plot is above the canopy. It can help to discuss the region where MOST (and therefore some of the models) is valid (surface layer).**

We have retained the z/h scaling since the convector effect is most prominent close to the canopy. In canopy turbulence studies where the canopy sub layer is discussed, the convention is using h, since the canopy sublayer is described as the region between h-5h. The plots start from z/h=1 to 10, which clearly means they are above the canopy- which would not be as clear if zi is used. Moreover, the reason why h is used for CSL is that is the most dominant length scales among a number of length scales in the canopy sub layer -so the choice is not arbitrary and has a physical basis.

**Figure 4: I believe that the blue captions below the figures are wrong.**

Thanks for pointing out. The blue captions are removed.